# Evaluating the impact of DREAMS on HIV incidence among adolescent girls and young women: A population-based cohort study in Kenya and South Africa

Isolde Birdthistle[1]*, Daniel Kwaro[2], Maryam Shahmanesh[3,4], Kathy Baisley[1,3], Sammy Khagayi[2], Natsayi Chimbindi[3], Vivienne Kamire[2], Nondumiso Mthiyane[3], Annabelle Gourlay[1], Jaco Dreyer[3], Penelope Phillips-Howard[5], Judith Glynn[1], Sian Floyd[1]

**1** Faculty of Epidemiology and Population Health, London School of Hygiene & Tropical Medicine, London, United Kingdom, **2** Centre for Global Health Research, Kenya Medical Research Institute, Kisumu, Kenya, **3** Africa Health Research Institute, Mtubatuba, KwaZulu-Natal, South Africa, **4** Institute for Global Health, University College London Institute of Child Health, London, United Kingdom, **5** Department of Clinical Sciences, Liverpool School of Tropical Medicine, Liverpool, United Kingdom

* Isolde.Birdthistle@lshtm.ac.uk

**Data Availability Statement:** The datasets used in this study have been deposited in data repositories and will be available upon request. Data from South

## Abstract

### Background

Through a multisectoral approach, the DREAMS Partnership aimed to reduce HIV incidence among adolescent girls and young women (AGYW) by 40% over 2 years in high-burden districts across sub-Saharan Africa. DREAMS promotes a combination package of evidence-based interventions to reduce individual, family, partner, and community-based drivers of young women's heightened HIV risk. We evaluated the impact of DREAMS on HIV incidence among AGYW and young men in 2 settings.

### Methods and findings

We directly estimated HIV incidence rates among open population-based cohorts participating in demographic and HIV serological surveys from 2006 to 2018 annually in uMkhanyakude (KwaZulu-Natal, South Africa) and over 6 rounds from 2010 to 2019 in Gem (Siaya, Kenya). We compared HIV incidence among AGYW aged 15 to 24 years before DREAMS and up to 3 years after DREAMS implementation began in 2016. We investigated the timing of any change in HIV incidence and whether the rate of any change accelerated during DREAMS implementation. Comparable analyses were also conducted for young men (20 to 29/34 years).

In uMkhanyakude, between 5,000 and 6,000 AGYW were eligible for the serological survey each year, an average of 85% were contacted, and consent rates varied from 37% to 67%. During 26,395 person-years (py), HIV incidence was lower during DREAMS implementation (2016 to 2018) than in the previous 5-year period among 15- to 19-year-old females (4.5 new infections per 100 py as compared with 2.8; age-adjusted rate ratio (aRR) = 0.62,

Africa are available from the AHRI data repository (https://data.ahri.org/index.php/home). Requests for the data from Kenya can be made to the KEMRI Scientific and Ethics Review Unit (SERU), at https://www.kemri.org/seru-overview or by contacting SERU at seru@kemri.org.

**Funding:** The impact evaluation of DREAMS is funded by the Bill and Melinda Gates Foundation (Grant OPP1136774 to IB, http://www.gatesfoundation.org). Foundation staff advised the study team, but did not substantively affect the study design, instruments, interpretation of data, or decision to publish. Africa Health Research Institute is supported by a grant from the Wellcome Trust (082384/Z/07/Z). The AHRI Demographic Surveillance Information System and Population Intervention Programme is funded by the Wellcome Trust (201433/Z/16/Z) and the South Africa Population Research Infrastructure Network (funded by the South African Department of Science and Technology and hosted by the South African Medical Research Council), with co-funding from the Bill and Melinda Gates Foundation. In Kenya, data were collected with funding from the President's Emergency Fund for AIDS Relief, under Cooperative Agreements with the US Centers for Disease Control and Prevention, and co-funding from the Bill & Melinda Gates Foundation. The funders had no role in study design, data collection and analysis, decision to publish, or preparation of the manuscript.

**Competing interests:** The authors have declared that no competing interests exist.

**Abbreviations:** YW, adolescent girls and young women; aRR, age-adjusted rate ratio; ART, antiretroviral treatment; CI, confidence interval; DREAMS, Determined, Resilient, Empowered, AIDS-free, Mentored, and Safe; DSS, demographic surveillance site; ELISA, enzyme-linked immunosorbent assay; PrEP, preexposure prophylaxis; py, person-years; RR, rate ratio; YWSS, young women who sell sex.

95% confidence interval [CI] 0.48 to 0.82), and lower among 20- to 24-year-olds (7.1/100 py as compared with 5.8; aRR = 0.82, 95% CI 0.65 to 1.04). Declines preceded DREAMS introduction, beginning from 2012 to 2013 among the younger and 2014 for the older women, with no evidence of more rapid decline during DREAMS implementation. In Gem, between 8,515 and 11,428 AGYW were eligible each survey round, an average of 34% were contacted and offered an HIV test, and consent rates ranged from 84% to 99%. During 10,382 py, declines in HIV incidence among 15- to 19-year-olds began before DREAMS and did not change after DREAMS introduction. Among 20- to 24-year-olds in Gem, HIV incidence estimates were lower during DREAMS implementation (0.64/100 py) compared with the pre-DREAMS period (0.94/100 py), with no statistical evidence of a decline (aRR = 0.69, 95% CI 0.53 to 2.18). Among young men, declines in HIV incidence were greater than those observed among AGYW and also began prior to DREAMS investments. Study limitations include low study power in Kenya and the introduction of other interventions such as universal treatment for HIV during the study period.

## Conclusions

Substantial declines in HIV incidence among AGYW were observed, but most began before DREAMS introduction and did not accelerate in the first 3 years of DREAMS implementation. Like the declines observed among young men, they are likely driven by earlier and ongoing investments in HIV testing and treatment. Longer-term implementation and evaluation are needed to assess the impact of such a complex HIV prevention intervention and to help accelerate reductions in HIV incidence among young women.

## Author summary

### Why was this study done?

- Adolescent girls and young women (AGYW) experience high risk of HIV infection relative to other demographic groups, and evidence is needed to drive down the rate of new infections.

- DREAMS is a large investment in combination HIV prevention for AGYW in 15 countries, and evidence of its effectiveness can guide ongoing efforts to achieve epidemic control and Sustainable Development Goal 3 to end AIDS as a public health threat by 2030.

### What did the researchers do and find?

- With large open cohorts of AGYW and young men resident in demographic surveillance sites in western Kenya and KwaZulu Natal, South Africa, we compared HIV incidence before DREAMS and up to 3 years after DREAMS implementation began.

- We investigated the timing of any change in HIV incidence and whether any decline in incidence rates accelerated during DREAMS implementation.

- In the first 3 years of DREAMS implementation, we did not observe an additional effect of DREAMS on HIV incidence reductions among AGYW.

- Declines in HIV incidence began before DREAMS rollout.

### What do these findings mean?

- The ongoing trend in HIV decline is most likely driven by investments in HIV testing, treatment, and male circumcision that preceded DREAMS introduction and continued alongside it.

- A complex, multisectoral programme like DREAMS needs time to scale up and strengthen linkages between social, structural, and biomedical interventions before it can yield measurable impacts on HIV incidence rates.

- With absolute levels of HIV risk remaining high for AGYW in uMkhanyakude, and relative risk higher among young females than young males in both settings, strengthening of HIV prevention is needed to accelerate HIV incidence declines.

## Introduction

Persistently high rates of HIV infection among adolescent girls and young women (AGYW) have led to large investments in targeted HIV prevention in eastern and southern Africa. This includes the "DREAMS Partnership" launched in 2015 by PEPFAR and private sector partners to promote "Determined, Resilient, Empowered, AIDS-free, Mentored, and Safe" (DREAMS) lives among AGYW [1]. Through a combination package of 12 evidence-based interventions, DREAMS promotes a multisectoral approach to reduce HIV incidence and the individual, family, and community-based drivers of young women's heightened risk. DREAMS was launched in 2015 and subsequently rolled out in 63 districts across 10 countries in East and Southern Africa. It was further expanded to new districts in 5 more countries in 2018, with successive budget increases as it evolved from central funding to country budgets (for example, US$189M for fiscal year 2020 and US$399M for 2021) [2].

A systematic review of directly observed HIV incidence estimates in the 10 DREAMS countries prior to DREAMS introduction showed that, while rates among AGYW have declined in many settings since rollout of antiretroviral treatment (ART; between 2005 and 2015), they remain substantially higher than those of their male peers in all settings [3]. The excessive risk among adolescent girls relative to males exposes a gender gap, which DREAMS aimed to address with urgency. Specifically, the aim of DREAMS was to reduce HIV incidence among AGYW by 40% over 2 years [1].

This is an ambitious goal, and one that is challenging to evaluate, since DREAMS districts were not selected at random but on the basis of HIV burden, and few DREAMS settings have pre-DREAMS estimates of HIV incidence to measure change over time. Longitudinal population-based surveillance of HIV is limited to about 10 settings in sub-Saharan Africa, and most are not places where DREAMS has been implemented [4]. We identified 2 settings in which historical HIV incidence data were collected in the general population prior to DREAMS introduction and could be compared to new data collected prospectively after DREAMS

rollout from early 2016 [5,6]. Here, we use retrospective and new data in both settings—districts with historically high prevalence and incidence of HIV in Kenya and South Africa—to investigate the impact of DREAMS among general populations of AGYW. We also tracked HIV incidence rates among young men in the typical age range for sexual partners of AGYW to understand trends in male risk and the epidemiological context for AGYW.

## Methods

### Data sources and collection, and study settings

As a complex, multisectoral programme addressing individual and contextual drivers of HIV risk, DREAMS should have both direct and indirect effects on HIV transmission. The impact of DREAMS on HIV incidence should therefore be evident at the population level. This population-based cohort study is reported as per STROBE guidelines (S1 STROBE Checklist).

To examine population-level time trends in HIV incidence, we used data from 2 community-wide demographic surveillance sites (DSS) in which DREAMS was implemented from 2016: uMkhanyakude, a rural district of KwaZulu-Natal in South Africa, and Gem, a sub-county of Siaya County in western Kenya. We used seroprevalence data collected before 2016 for baseline (pre-DREAMS) measures. Serological data were collected annually since 2006 in uMkhanyakude, and over 3 rounds during 2010 to 2014 in Gem [7,8]. In both settings, further prospective rounds of HIV serosurveillance were conducted annually between 2016 and 2019 and used to estimate HIV incidence in that time period. The methods and settings are described in detail elsewhere [5,7–9]. Surveillance procedures included collection of a dried blood spot for anonymised testing in uMkhanyakude, while in Gem, home-based rapid HIV testing was provided as a service and a self-report of HIV-positive status was also taken as evidence of an individual's HIV status.

In South Africa, where pre-DREAMS annual HIV incidence was estimated to be 6% among 15- to 24-year-old AGYW, we estimated that there would be at least 3,000 person-years (py) of follow-up after DREAMS rollout and that study power was 95% to show a 40% reduction in HIV incidence. In Kenya, where pre-DREAMS HIV incidence was much lower, at 0.7% among AGYW aged 15 to 24 years, we estimated that with 9,000 py of follow-up, study power was 70% to show a 40% reduction in HIV incidence [5]. To maximise participation of AGYW in post-DREAMS surveys, we intensified recruitment efforts with dedicated tracking teams, point-of-care HIV testing available to those wishing to know their HIV status, and compensation of mobile phone airtime from 2017 (in uMkhanyakude; 10 Rand (US$0.70) provided to the 15- to 24-year-old living in the household), and greater frequency of survey rounds (in Gem).

### Statistical analysis

Analyses were restricted to residents aged 15 to 24 years who had at least 2 HIV test results, with the first HIV-negative test before the age of 25. HIV incidence rates were calculated as the number of seroconversions per 100 py of observation. Participants entered the analysis from their first HIV negative test and exited with the latest of date of the last negative test or their estimated seroconversion date, if that date was before age 25 years. Individuals whose last negative test/estimated seroconversion was after the age of 25 were censored when they turned 25. We compared the age distribution of the eligible AGYW who did and did not contribute person-years of follow-up to the HIV incidence analyses in each calendar year (those with and without repeat HIV tests during or after the calendar year being considered).

Seroconversion dates were multiply-imputed (a minimum of 100 imputations was prespecified; 100 imputations were used in Gem and 250 in uMkhanyakude) as a fraction of the

interval between the last negative test date and the first positive test date, assuming a uniform distribution. This was done in order to avoid grouping (clustering) of seroconversion dates by calendar year, which will occur with the simpler "mid-point" imputation method unless sero-surveys are conducted annually. Some participants leave the surveillance areas for a period of time and later return. To exclude from the analysis the infections that may have occurred outside the DREAMS intervention areas, we excluded seroconversions for which the imputed seroconversion date was during a period when an individual was resident outside the area and censored those participants (as HIV negative) on the date of out-migration before the seroconversion [7]. Correspondingly, among individuals who remained HIV negative, we also excluded from the analysis the person-years during which they were not resident in the DSS area. As a sensitivity analysis, HIV incidence rates were also calculated with the inclusion of all nonresidency periods, in which all seroconversions contributed to the numerator and all person-time during periods of nonresidency contributed to the denominator.

The *a priori* analysis plan was to compare HIV incidence among AGYW in calendar periods after DREAMS was rolled out with the 5-year period immediately prior to rollout (2011 to 2015) [5]. The DREAMS scale-up period was monitored up to 3 years after DREAMS interventions were introduced, that is, 2016 to 2019. (DREAMS interventions were funded through 2018 in uMkhanyakude and 2019 in Gem.) In Gem, we included all the 2019 serological data to estimate HIV incidence rates to the end of 2019. In uMkhanyakude, we used the 2019 serological data but censored the analysis at the end of 2018, because the 2019 surveillance data on residency were incomplete and because DREAMS investments stopped in this setting by late 2018.

Poisson regression was used to estimate rate ratios (RRs) and 95% confidence intervals (CIs) for the effect of calendar period on HIV incidence, overall and separately by age group (age 15 to 19 years and 20 to 24 years). In uMkhanyakude, where serological surveys were conducted annually, we also estimated HIV incidence for every calendar year and used RRs to compare rates from one year to the next to identify the timing of any change in trend. Informed by these time trends in annual incidence rates, we created and compared *a posteriori* calendar periods for evidence of additional change in HIV incidence rates during the DREAMS scale-up period. This approach fits within the framework of an interrupted time series analysis as it identifies a key time point, and then asks whether a time trend alters the trajectory (slope) after that time point [10]. In Gem, where pre-DREAMS surveys were less frequent, the 2010 to 2015 baseline period was split into 2 separate periods (2010 to 2012 and 2013 to 2015), *a posteriori*, to assess the timing of changes in HIV incidence rates more precisely, controlling for age group.

HIV incidence trends were also estimated for young men characterised in earlier research to be in the typical age range for male sexual partners of AGYW: 20 to 29 years in uMkhanyakude and 20 to 34 years in Gem [11]. We used the same analytical methods to estimate HIV incidence by calendar time period, as described above for AGYW, censoring data at the estimated time of seroconversion or by the age of 29 in uMkhanyakude and 34 years in Gem.

## Laboratory methods

In uMkhanyakude, South Africa, dried blood spots were tested for HIV antibody using an enzyme-linked immunosorbent assay (ELISA). From 2006 to 2008, Vironostika HIV Uni-Form II Ag/Ab (bioMérieux, Boxtel, The Netherlands) was used, with the GAC ELISA (Bio-Rad, Marnes-la-Coquette, France) as a confirmatory test. From 2008 to 2019, the SD HIV 1/2 ELISA (V3) was used. In Gem, Kenya, rapid HIV testing followed the national testing algorithm of the Kenyan Ministry of Health. Samples were tested with Determine (Alere, Orlando, FL, USA), with positive tests confirmed with SD Bioline HIV-1/2 3.0 in 2011 and 2012 (Standard

Diagnostics, Giheung-gu, Gyonggi-do, South Korea) and First Response (Premier Medical Corporation, Nadi Daman, India) in 2016, and Uni-gold (Trinity Biotech, Bray, Wicklow, Ireland) used as a referee test. In both settings, the DSS resident identification number was recorded and used for linkage of individuals' HIV test results and data from questionnaires.

### Patient and public involvement, and ethics approvals

In both settings, studies are presented to the Community Advisory Board for their input and permission before submitting for ethics clearance. The study protocol and tools were approved by ethics committees at the London School of Hygiene and Tropical Medicine and Liverpool School of Tropical Medicine in the United Kingdom; Kenya Medical Research Institute in Kenya; and University of KwaZulu-Natal in South Africa. Participants provided informed, written consent to participate as did parents/guardians of legal minors under age 18 years.

### Results

In uMkhanyakude, from 2006 to 2019, between 5,000 and 6,000 AGYW aged 15 to 24 years were eligible for the serological survey each year, and an average of 85% were contacted (range 76% to 96%) (Table 1). Consent rates were under 50% from 2006 to 2014, after which annual participation rates (among those contacted) rose to around 60%, with intensified recruitment and tracking of AGYW from 2017. Around 3,000 AGYW were eligible for the HIV incidence cohort each year (first observed as HIV negative before age 25, and without evidence of seroconversion before that year), of whom an average of 75% had a repeat HIV test in that year or a later one and so contributed person-time to the analysis. Total follow-up time was 26,393 py: 14,184 py among 15- to 19-year-olds; and 12,209 py among 20- to 24-year-olds. For each calendar year of follow-up, AGYW who contributed person-years to the analysis were on average

**Table 1. Testing participation in HIV serosurveys among AGYW and contribution to HIV incidence cohort in uMkhanyakude, South Africa.**

| Year | HIV survey | | | HIV incidence cohort | |
|------|-----------|---|---|---------------------|---|
| | Eligible[1] | Contacted (% of eligible)[1] | Consented (% of contacted)[1] | Eligible (aged <25 years)[2] | Repeat testers[3] |
| 2006 | 5,822 | 5,558 (95.5%) | 2,578 (46.4%) | 3,246 | 2,700 (83.2%) |
| 2007 | 6,033 | 5,396 (89.4%) | 2,165 (40.1%) | 3,268 | 2,595 (79.4%) |
| 2008 | 5,937 | 5,443 (91.7%) | 2,057 (37.8%) | 3,103 | 2,569 (82.8%) |
| 2009 | 4,977 | 4,449 (89.4%) | 1,623 (36.5%) | 2,763 | 2,232 (80.8%) |
| 2010 | 5,743 | 4,768 (83.0%) | 2,154 (45.2%) | 3,018 | 2,383 (79.0%) |
| 2011 | 4,959 | 4,278 (86.3%) | 1,868 (43.7%) | 2,791 | 2,214 (79.3%) |
| 2012 | 5,406 | 4,207 (77.8%) | 1,582 (37.6%) | 2,757 | 2,070 (75.1%) |
| 2013 | 5,534 | 4,428 (80.0%) | 2,082 (47.0%) | 3,000 | 2,248 (74.9%) |
| 2014 | 5,390 | 4,265 (79.1%) | 1,878 (44.0%) | 2,987 | 2,264 (75.8%) |
| 2015 | 5,039 | 4,367 (86.7%) | 2,613 (59.8%) | 3,087 | 2,426 (78.6%) |
| 2016 | 5,249 | 4,509 (85.9%) | 3,020 (67.0%) | 3,395 | 2,525 (74.4%) |
| 2017 | 5,220 | 4,404 (84.4%) | 2,088 (47.4%) | 3,180 | 2,194 (69.0%) |
| 2018 | 5,029 | 3,834 (76.2%) | 2,345 (61.2%) | 3,153 | 1,809 (57.4%) |

[1]Among AGYW aged 15–24 years.

[2]Cumulative number of AGYW who first tested HIV negative when aged <25 years are eligible for entry into the HIV incidence cohort and are still aged <25 years.

[3]Number of eligible HIV negative AGYW who had a repeat test and contributed person-time to the incidence analysis during each calendar period. For example, if an AGYW first tests HIV negative in 2006 and has a second HIV-negative test in 2010, she contributes person time to 2006, 2007, 2008, 2009, and 2010. Individuals leave the AGYW incidence cohort at the earliest estimate of seroconversion or reaching the age of 25.

AGYW, adolescent girls and young women.

**Table 2. Testing participation in HIV serosurveys among AGYW and contribution to HIV incidence cohort in uMkhanyakude, South Africa (A) and Gem, Kenya (B).**

| Round | HIV survey | | | HIV incidence cohort | |
|---|---|---|---|---|---|
| | Eligible for HIV survey[1] | Contacted and offered HIV test (% of eligible) | Consented[2] (% of offered) | Eligible for HIV incidence cohort[3] | HIV results available[4] |
| 2010–2012 [10/2010–10/2012] | 11,428 | 6,041 (52.9%) | 6,018 (99.6%) | 5,707 | 3,386 (59.3%) |
| 2013–2014 [06/2013–08/2014] | 10,549 | 2,839 (26.9%) | 2,757 (97.1%) | 5,612 | 2,290 (40.8%) |
| 2016 [05/2016–09/2016] | 8,515 | 1,692 (19.9%) | 1,640 (96.9%) | 4,171 | 2,106 (50.5%) |
| 2017 [01/2017–09/2017] | 9,432 | Unknown* | 1,319 | 4,273 | 1,700 (39.8%) |
| 2018 [01/2018–01/2019] | 10,299 | 3,706 (36.0%) | 3,435 (92.7%) | 5,892 | 2,323 (39.4%) |
| 2019 [01/2019–11/2019] | 9,920 | 3,438 (34.7%) | 2,900 (84.4%) | 5,890 | 41 (0.7%) |

[1]AGYW aged 15–24 years who were resident in a randomly selected study compound during the serosurvey round.

[2]Consented to participate (to provide a blood sample for HIV testing or self-reported as HIV+).

[3]Cumulative number of AGYW who first tested HIV negative when aged <25 and are eligible for entry into the HIV incidence cohort and are still aged <25 years.

[4]Number of eligible HIV negative AGYW who had a repeat test and contributed person-time to the HIV incidence analysis during each calendar period.

*Those who were NOT offered and those who did NOT consent were not collected during the 2017 survey round.

AGYW, adolescent girls and young women.

slightly older than those who did not contribute; the differences in mean age were small (<1 year) and changed little over the study period (S1 Table).

In Gem, 2 rounds of serological surveys were conducted before DREAMS introduction, between 2010 and 2015, and annual rounds were subsequently conducted between 2016 and 2019 (Table 2). Between 8,515 and 11,428 AGYW aged 15 to 24 years were eligible each round, and an average of 34% were contacted and offered an HIV test. Consent rates ranged between 84% to 99%. Of those eligible for the HIV incidence cohort each year, between 40% and 60% had a repeat test and contributed person-time to the HIV incidence analysis. AGYW with repeat HIV test results contributed a total of 10,382 py to the HIV incidence cohort (6,102 py among 15- to 19-year-olds; 4,279 py among 20- to 24-year-olds), with 2,325 py of follow-up during the DREAMS implementation period 2016 to 2019.

## HIV incidence trends among AGYW, by calendar period and round

In the calendar period since DREAMS implementation began in uMkhanyakude (2016 to 2018), rates of HIV incidence among AGYW were lower, overall and in both age groups, compared to the 5-year period immediately prior to DREAMS. Rates were lower by 38% among 15- to 19-year-olds in that time frame (4.5 cases per 100 py in 2011 to 2015 as compared with 2.8 per 100 py in 2016 to 2018 [age-adjusted RR (aRR) = 0.62; 95% CI 0.48 to 0.82]), and by 18% among 20- to 24-year-olds (7.1 per 100 py as compared with 5.8 [aRR = 0.82; 95% CI 0.65 to 1.04]) (Table 3). Estimates of HIV incidence calculated for each individual year indicate that annual declines in HIV incidence began before DREAMS introduction in 2016 (Table 4,

**Table 3. Incidence of HIV infection among AGYW, by age and DREAMS implementation period in uMkhanyakude, South Africa and Gem, Kenya.**

| Age group | Calendar period | New HIV infections | Person-years | Incidence rate/100 py | aRR (95% CI)[1] |
|---|---|---|---|---|---|
| *uMkhanyakude, South Africa* | | | | | |
| 15–19 years | 2006–2010 | 269 | 5,663 | 4.75 (4.13–5.46) | 1.04 (0.85–1.28) |
| | 2011–2015 | 227 | 4,992 | 4.54 (3.93–5.25) | 1 |
| | 2016–2018 | 98 | 3,529 | 2.78 (2.24–3.46) | 0.62 (0.48–0.82) |
| 20–24 years | 2006–2010 | 365 | 4,866 | 7.50 (6.64–8.47) | 1.06 (0.89–1.27) |
| | 2011–2015 | 336 | 4,743 | 7.08 (6.23–8.04) | 1 |
| | 2016–2018 | 151 | 2,600 | 5.80 (4.81–7.00) | 0.82 (0.65–1.04) |
| 15–24 years | 2006–2010 | 634 | 10,529 | 6.02 (5.51–6.58) | 1.06 (0.92–1.21) |
| | 2011–2015 | 563 | 9,736 | 5.78 (5.27–6.34) | 1 |
| | 2016–2018 | 249 | 6,130 | 4.07 (3.54–4.67) | 0.74 (0.62–0.87) |
| *Gem, Kenya* | | | | | |
| *- Comparison of 2 calendar periods (a priori analysis)* | | | | | |
| 15–19 years | 2010–2015 | 30 | 4,866 | 0.62 (0.40–0.94) | 1 |
| | 2016–2019 | 6 | 1,236 | 0.49 (0.16–1.50) | 0.79 (0.24–2.63) |
| 20–24 years | 2010–2015 | 31 | 3,191 | 0.97 (0.63–1.50) | 1 |
| | 2016–2019 | 7 | 1,088 | 0.64 (0.28–1.49) | 0.66 (0.26–1.70) |
| 15–24 years | 2010–2015 | 61 | 8,057 | 0.76 (0.55–1.04) | 1 |
| | 2016–2019 | 13 | 2,325 | 0.56 (0.28–1.11) | 0.74 (0.36–1.53) |
| *- Comparison of 3 calendar periods (a posteriori analysis)* | | | | | |
| 15–19 years | 2010–2012 | 20 | 2,171 | 0.92 (0.57–1.49) | 2.48 (1.16–5.31) |
| | 2013–2015 | 10 | 2,695 | 0.37 (1.90–7.30) | 1 |
| | 2016–2019 | 6 | 1,236 | 0.49 (0.16–1.50) | 1.31 (0.35–4.89) |
| 20–24 years | 2010–2012 | 16 | 1,588 | 1.01 (0.60–1.69) | 1.08 (0.53–2.18) |
| | 2013–2015 | 15 | 1,603 | 0.94 (0.52–1.70) | 1 |
| | 2016–2019 | 7 | 1,088 | 0.64 (0.28–1.49) | 0.69 (0.53–2.18) |
| 15–24 years | 2010–2012 | 36 | 3,759 | 0.96 (0.67–1.36) | 1.65 (0.99–2.75) |
| | 2013–2015 | 25 | 4,298 | 0.58 (0.37–0.92) | 1 |
| | 2016–2019 | 13 | 2,325 | 0.56 (0.28–1.11) | 0.96 (0.43–2.14) |

[1]Adjusted for current age.

AGYW, adolescent girls and young women; aRR, age-adjusted rate ratio; CI, confidence interval; py, person-years.

Fig 1). Rates among 15- to 19-year-olds were highest in 2012, after which they subsequently declined each year through 2017, with the difference compared to 2011 widening with each successive year. Among 20- to 24-year-olds, declines began later, from 7.8 cases per 100 py in 2014 followed by annual reductions thereafter to 5.2 per 100 py in 2018.

In comparisons of *a posteriori* calendar periods informed by the observed trends in annual incidence rates, among the younger cohort, we found evidence that HIV incidence fell during 2013 to 2015 by around 13% per year (aRR = 0.87; 95% CI 0.76 to 0.99, $p$ = 0.03) (Table 5). During the 3 years, 2016 to 2018, this rate of reduction continued, but there was no evidence of additional change (aRR for additional annual rate of change during 2016 to 2018 = 1.06; 95% CI 0.81 to 1.39). Among the older cohort, there was no evidence of decline between 2006 and 2014, a suggestion of a decline of around 9% comparing 2015 with 2014 (aRR = 0.91, 95% CI [0.63 to 1.31]), and no evidence of more rapid decline during 2016 to 2018 (aRR for additional annual rate of change during 2016 to 2018 = 1.02, 95% CI 0.63 to 1.64). Combining the years 2015 to 2018, there was weak evidence that HIV incidence fell by around 8% per year over this 4-year period (aRR = 0.92; 95% CI 0.84 to 1.01; $p$ = 0.07).

**Table 4. Incidence of HIV infection among AGYW, by age and individual year in uMkhanyakude (2006–2018).**

| Age group | Year | New HIV infections | Person-years | Incidence rate/100 py | RR (95% CI) (reference year 2011) |
|---|---|---|---|---|---|
| **15–19 years** | 2006 | 61 | 1,314 | 4.60 (3.30–6.42) | 0.93 (0.56–1.53) |
| | 2007 | 57 | 1,252 | 4.53 (3.25–6.32) | 0.92 (0.55–1.53) |
| | 2008 | 55 | 1,197 | 4.55 (3.23–6.41) | 0.92 (0.55–1.54) |
| | 2009 | 52 | 1,006 | 5.14 (3.62–7.32) | 1.04 (0.62–1.75) |
| | 2010 | 44 | 893 | 4.91 (3.32–7.27) | 0.99 (0.56–1.76) |
| | 2011 | 47 | 951 | 4.95 (3.38–7.23) | 1 |
| | 2012 | 47 | 855 | 5.45 (3.77–7.87) | 1.10 (0.63–1.92) |
| | 2013 | 48 | 969 | 4.96 (3.45–7.14) | 1.00 (0.58–1.73) |
| | 2014 | 44 | 1,073 | 4.08 (2.77–6.02) | 0.83 (0.48–1.41) |
| | 2015 | 40 | 1,145 | 3.48 (2.37–5.11) | 0.70 (0.40–1.22) |
| | 2016 | 35 | 1,240 | 2.80 (1.84–4.24) | 0.57 (0.32–0.99) |
| | 2017 | 30 | 1,211 | 2.48 (1.56–3.94) | 0.50 (0.28–0.90) |
| | 2018 | 33 | 1,078 | 3.04 (2.01–4.59) | 0.61 (0.35–1.07) |
| **20–24 years** | 2006 | 70 | 880 | 7.96 (5.92–10.72) | 1.20 (0.77–1.88) |
| | 2007 | 71 | 945 | 7.42 (5.44–10.12) | 1.12 (0.71–1.77) |
| | 2008 | 74 | 1,001 | 7.33 (5.45–9.84) | 1.11 (0.70–1.74) |
| | 2009 | 76 | 1,023 | 7.35 (5.46–9.90) | 1.11 (0.70–1.76) |
| | 2010 | 75 | 1,017 | 7.34 (5.44–9.90) | 1.11 (0.69–1.77) |
| | 2011 | 66 | 994 | 6.62 (4.74–9.25) | 1 |
| | 2012 | 69 | 961 | 7.18 (5.31–9.70) | 1.08 (0.68–1.74) |
| | 2013 | 67 | 942 | 7.04 (5.13–9.66) | 1.06 (0.67–1.69) |
| | 2014 | 73 | 934 | 7.74 (5.79–10.36) | 1.17 (0.75–1.82) |
| | 2015 | 61 | 911 | 6.67 (4.86–9.16) | 1.01 (0.63–1.61) |
| | 2016 | 54 | 854 | 6.27 (4.38–8.95) | 0.95 (0.58–1.54) |
| | 2017 | 52 | 889 | 5.83 (4.09–8.30) | 0.88 (0.54–1.43) |
| | 2018 | 45 | 858 | 5.22 (3.59–7.60) | 0.79 (0.48–1.31) |

AGYW, adolescent girls and young women; CI, confidence interval; py, person-years; RR, rate ratio.

In Gem, HIV incidence estimates during the DREAMS implementation period (2016 to 2019) were lower than those observed in the 5-year period prior to DREAMS (2010 to 2015), but there was considerable statistical uncertainty around the estimates (Table 3). For example, among 15- to 24-year-olds overall, HIV incidence was estimated as 0.56 cases per 100 py (95% CI 0.28 to 1.11) since DREAMS began, as compared with 0.76 per 100 py (95% CI 0.55 to 1.04) in the baseline period (aRR = 0.74, 95% CI 0.35 to 1.53).

The incidence rate in Gem was highest in the earlier surveillance period (2010 to 2012) at 0.96% (95% CI 0.67 to 1.36) and declined to 0.58% (95% CI 0.37 to 0.92) by 2013 to 2015 (aRR = 1.65, 95% CI 0.99 to 2.75) (Table 3). This difference was driven by a reduction in rates among the younger cohort (15- to 19-year-olds). In *a posteriori* analysis, defining the baseline period as 2013 to 2015, we found no evidence of a reduction among 15- to 24-year-olds overall since DREAMS rollout in Gem (aRR = 0.96, 95% CI 0.43 to 2.14) (Table 3, Fig 2). Compared to 2013 to 2015, there was no evidence of a decline in HIV incidence during the DREAMS implementation period in the 15- to 19-year-olds (aRR 1.31, 95% CI 0.35 to 4.89) or among the 20- to 24-year-olds (aRR = 0.69, 95% CI 0.53 to 2.18).

Sensitivity analyses estimating HIV incidence regardless of gaps in residency (that is, including seroconversions and follow-up time spent outside of the geographic surveillance area of Gem) enabled the inclusion of substantially more person-years of follow-up post-

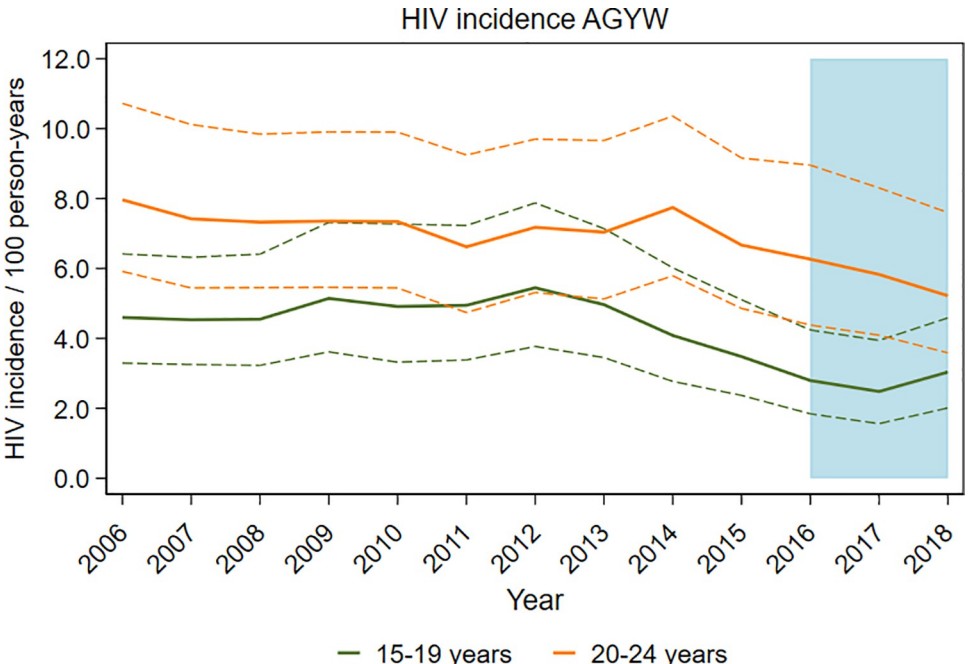

**Fig 1. HIV incidence rates over time among AGYW with 95% CIs, in uMkhanyakude, South Africa.** AGYW, adolescent girls and young women; CI, confidence interval.

DREAMS introduction in Gem (with a total of 8,456 py among 15- to 24-year-old women; S2 Table). This analysis generated lower estimates of HIV incidence across age groups and time but yielded similar time trends. That is, declines in HIV incidence among the younger cohort were seen before DREAMS implementation (between 2010 and 2012 and 2013 and 2015) but not thereafter. Among the older cohort, there was no evidence of decline before DREAMS, and a 25% reduction observed during DREAMS implementation that did not achieve statistical significance (aRR = 0.75, 95% CI 0.43 to 1.29).

**Table 5. Trends in HIV incidence among AGYW, by age and _a posteriori_ calendar time periods in uMkhanyakude, South Africa.**

| Age group | Comparison* | Age-adjusted linear RR (95% CI) | _P_ value |
|---|---|---|---|
| 15–19 years | Change in 2011–2012 vs 2006–2010 | 1.12 (0.86–1.45) | 0.41 |
| | Trend from 2013 to 2015 | 0.87 (0.76–0.99) | 0.03 |
| | Additional change from 2016 to 2018 | 1.06 (0.81–1.39) | 0.65 |
| 20–24 years | Trend from 2006 to 2010 | 0.97 (0.89–1.05) | 0.48 |
| | Trend from 2011 to 2014 | 1.01 (0.93–1.10) | 0.74 |
| | Change in 2015 vs 2014 | 0.91 (0.63–1.31) | 0.61 |
| | Additional change from 2016 to 2018 | 1.02 (0.63–1.64) | 0.94 |
| 20–24 years | Trend from 2006 to 2010 | 0.97 (0.90–1.05) | 0.48 |
| | Trend from 2011 to 2014 | 1.01 (0.94–1.09) | 0.73 |
| | Trend from 2015 to 2018 | 0.92 (0.84–1.01) | 0.07 |

*Time periods are informed by the year in which HIV incidence first began to decline, for 15- to 19-year-olds and 20- to 24-year-olds, based on Table 4.

AGYW, adolescent girls and young women; CI, confidence interval; py, person-years; RR, rate ratio.

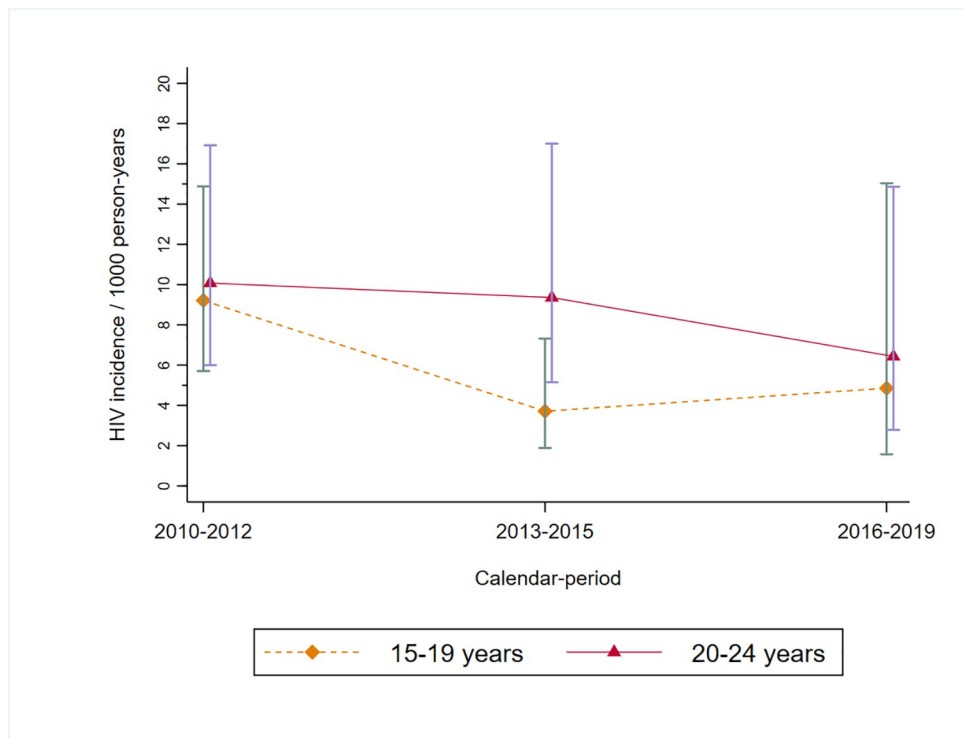

**Fig 2. HIV incidence rates over time among AGYW with 95% CIs, in Gem, Kenya.** AGYW, adolescent girls and young women; CI, confidence interval.

### HIV incidence trends among young men

Among young men aged 20 to 29 years in uMkhanyakude, lower rates of HIV incidence were observed in the DREAMS implementation period: 1.1% in 2016 to 2018 as compared with 2.6% in the preceding 5 years among those aged 20 to 24 (aRR = 0.44, 95% CI 0.26 to 0.77); and 2.4% versus 4.1% among men aged 25 to 29 years (aRR = 0.58, 95% CI 0.34 to 0.99) (Table 6). As with AGYW in this setting, estimates calculated for each annual round show that declines began before DREAMS introduction, from about 2012 when incidence peaked at 3.3% among 20- to 24-year-old men and from 2013 when rates among 25- to 29-year-olds were highest at 4.7%. There were steady declines each year thereafter through 2018, in both age groups (S3 Table, Fig 3).

Among young men aged 20 to 34 years in Gem, HIV incidence rates were lower with each successive calendar period, with uncertainty around the estimates (Table 6, Fig 4). In sensitivity analyses estimating HIV incidence without residency gaps and thus including substantially more person-time, there was no evidence of change in HIV incidence over time (aRR = 0.94, 95% CI 0.32 to 2.80) (S4 Table).

### Discussion

We observed large declines in HIV incidence among AGYW in both settings over the duration of this study. With data from HIV surveillance rounds conducted between 2010 and 2015 and prospective serological surveys conducted in the first 3 years of DREAMS implementation, we found that infection rates were lower in the DREAMS implementation period than the preceding 5-year calendar period. However, in more detailed analysis of trends, with annual

**Table 6. Incidence of HIV infection among young men, by age and DREAMS scale-up period in uMkhanyakude, South Africa and Gem, Kenya.**

| Age group | Calendar period | New HIV infections | Person-years | Incidence rate/100 py | aRR (95% CI)[1] |
|---|---|---|---|---|---|
| ***uMkhanyakude, South Africa*** | | | | | |
| 20–24 years | 2006–2010 | 119 | 3,901 | 3.04 (2.43–3.79) | 1.19 (0.86–1.65) |
| | 2011–2015 | 99 | 3,865 | 2.56 (2.03–3.22) | 1 |
| | 2016–2018 | 24 | 2,081 | 1.13 (0.70–1.83) | 0.44 (0.26–0.77) |
| 25–29 years | 2006–2010 | 67 | 1,507 | 4.41 (3.28–5.92) | 1.08 (0.73–1.60) |
| | 2011–2015 | 97 | 2,375 | 4.07 (3.18–5.20) | 1 |
| | 2016–2018 | 30 | 1,262 | 2.36 (1.49–3.73) | 0.58 (0.34–0.99) |
| ***Gem, Kenya*** | | | | | |
| 20–24 years | 2010–2012 | 10 | 1,421 | 0.70 (0.38–1.31) | 2.30 (0.73–7.26) |
| | 2013–2015 | 6 | 1,957 | 0.31 (0.11–0.86) | 1 |
| | 2016–2019 | 3 | 1,462 | 0.21 (0.07–0.64) | 0.67 (0.14–3.10) |
| 25–34 years | 2010–2012 | 21 | 1,940 | 1.08 (0.71–1.66) | 1.54 (0.79–2.99) |
| | 2013–2015 | 16 | 2,277 | 0.70 (0.41–1.21) | 1 |
| | 2016–2019 | 8 | 1,376 | 0.58 (0.29–1.16) | 0.83 (0.38–1.80) |

[1]Adjusted for current age.

aRR, age-adjusted rate ratio; CI, confidence interval; py, person-years.

surveillance data in uMkhanyakude and comparison of *a posteriori* calendar periods in both settings, we conclude that declines began several years before DREAMS introduction and continued, but did not accelerate, during DREAMS first 3 years of implementation.

Among young women aged 20 to 24 years in Gem, HIV incidence remained stable prior to DREAMS introduction (at approximately 1% between 2010 and 2015) and declined by 31%

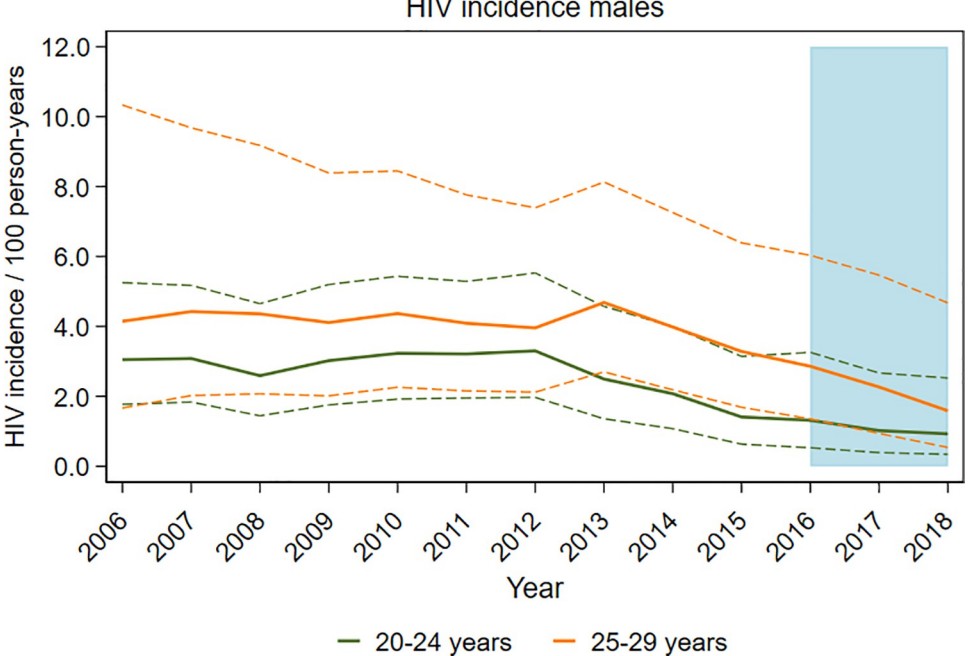

**Fig 3. HIV incidence rates over time among young men with 95% CIs, in uMkhanyakude, South Africa.** CI, confidence interval.

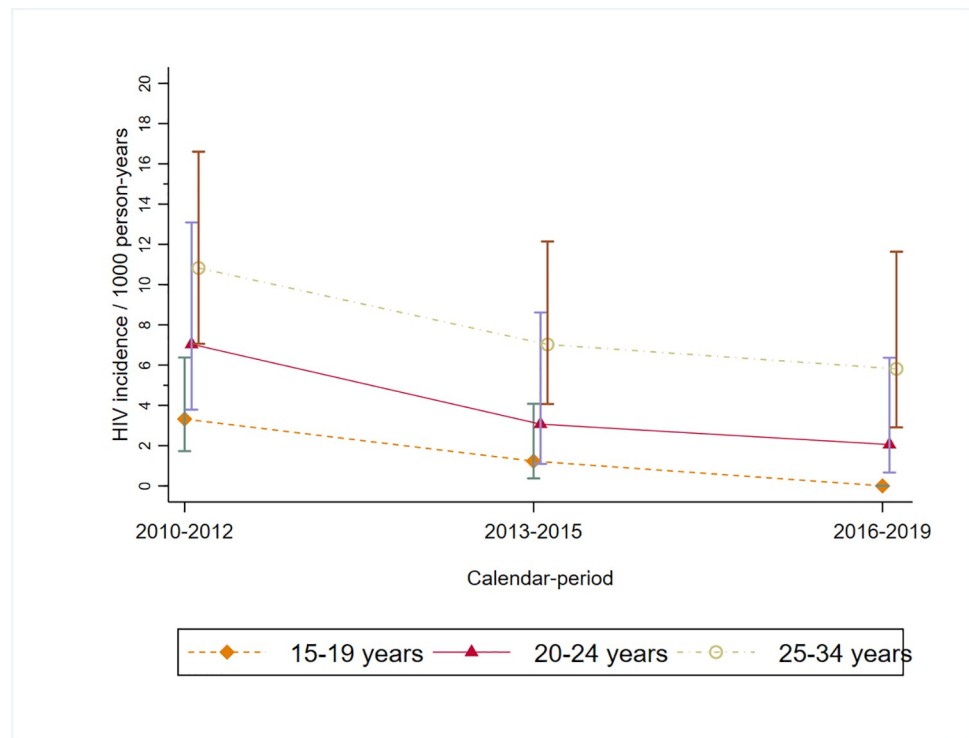

**Fig 4. HIV incidence rates over time among young men with 95% CIs, in Gem, Kenya.** CI, confidence interval.

during the DREAMS implementation period 2016 to 2019; however, statistical evidence (a wide confidence interval, indicating that the data were compatible with an RR of 1), shows that this may be a chance finding. Despite this uncertainty, it is plausible that DREAMS reduced HIV risk among young women in Gem, where the same evaluation found that DREAMS had a high reach and good layering of multiple interventions that intensified with time [12]. As part of the same evaluation, we established and followed nested closed cohorts of AGYW, randomly sampled from the population platform and followed over 2 years, to measure DREAMS coverage and causal impacts on individual-level outcomes [5,12]. In Gem, we observed a reduction in lifetime number of sexual partners and condomless sex among DREAMS beneficiaries compared with nonbeneficiaries, and evidence of other changes along a causal pathway between DREAMS and HIV risk, such as knowledge of HIV status and social support [13–15]. There were weaknesses, however, in the HIV "prevention cascade"—with poor links from high HIV testing to prevention services like preexposure prophylaxis (PrEP) and condom promotion, and uptake of both remained low by 2019. With more time and availability of PrEP, these linkages could be strengthened to improve young women's prevention choices in this setting [6,14,16].

The absence of an impact on HIV incidence in uMkhanyakude is consistent with other evidence emerging in that setting, including no evidence of DREAMS impact on the high rates of HSV-2 among AGYW [17]. Individual-level analyses comparing outcomes among DREAMS beneficiaries and nonbeneficiaries suggest that DREAMS did not directly affect behavioural drivers of sexual risk in this setting. Like in Gem, reach of DREAMS interventions from 2016 was high among the general population of AGYW in uMkhanyakude, but investment ceased by 2018, before PrEP for HIV was integrated within DREAMS and arguably before normative changes could occur [12,18,19]. Also, deeper engagement with young women and the wider

community could have strengthened the acceptability, innovation, and adaptation of DREAMS interventions to the needs and challenges of this particular context [18]. The HIV declines may be due to time trends in population- and partner-level risks, which began before DREAMS and continued afterwards. For example, wide-scale rollout of HIV testing, treatment, and voluntary male circumcision services have been shown to increase coverage against 90:90:90 targets among young men, thus reducing levels of untreated HIV infection among men in the typical age bracket for sexual partners of AGYW [20,21]. Modelling and phylogenetic studies suggest that preventing transmission from one young man can avert infections in up to 4 young women [22,23]. The significant reductions we observed in HIV incidence among young men since 2012 to 2013, together with increases in reported condom use, circumcision, and HIV treatment uptake, would have altered the context of HIV risk for AGYW prior to DREAMS investments in uMkhanyakude [24].

These data contribute to growing evidence that HIV incidence has reduced for AGYW across east and southern Africa, in the years leading up to 2020, though they remain at very high risk relative to men and other age groups [3]. In both study settings, HIV incidence declined first among the younger cohort (circa 2013 for adolescent girls aged 15 to 19 years) and later among young women aged 20 to 24 years—the age when HIV incidence peaked for women. Declines have also taken longer to occur in very high prevalence settings like uMkhanyakude, compared to lower-prevalence settings [25]. Despite the encouraging declines observed with time, the most recent estimates remain very high at >5% incidence among women aged 20 to 24 years in uMkhanyakude, about 5 times higher than men of the same age (approximately 1% in 2017). The estimated 3% incidence among girls aged 15 to 19 years in 2018 shows that absolute levels of risk are still high from an early age, despite steady reductions since 2012. High rates have also been reported in other settings of KwaZulu Natal, for example, in Durban, where the HIV epidemic has been described as "continuous, unrelenting, hyper" [26].

In such settings, in which HIV and other STI risks (and HIV/STI coinfection) remain high, the indirect effects of ART availability and male partner risk are slow to change the context of risk for young women and more must be done to protect young women from infection. Even community-randomised trials of population-wide universal test and treat interventions, offered intensively in high-burden settings, have yielded modest reductions in HIV incidence over a 3- to 4-year time frame, of the order of 20% to 30% [20,27]. DREAMS sought to address the drivers of AGYW risk directly and indirectly, comprehensively, and urgently, but investments in uMkhanyakude stopped in 2018, before the programme had an opportunity to embed in a high-priority setting [18,28]. In this setting, and in a related impact evaluation in Zimbabwe, very low proportions of young women who sell sex (YWSS) were reached with DREAMS interventions, limiting plausibility of its impact among the highest-risk women [19,29]. Research to understand the profile of AGYW who were reached by DREAMS in the Kenyan and South African study settings revealed that those who may be at most immediate sexual risk (for example, sexually active, ever pregnant) were less likely to be invited to DREAMS, although this improved with time [12]. DREAMS set a very ambitious target in a short time frame of 2 years. With more time to scale up and adapt interventions to reach those who need them most, by addressing stigma towards YWSS and training implementers be more inclusive, integrated packages that blend biomedical (including PrEP) and structural interventions (to boost education, employment, and empowerment) can be crucial to reduce young women's risk. Meanwhile, efforts to reduce male partner risk should also be intensified even in contexts of high treatment coverage, by targeting young men for earlier HIV diagnosis, treatment, and VMMC to avoid transmission in early stages of infection [20,23,30].

## Strengths and limitations

A strength of the study lies in the directly observed measure of HIV incidence, at a representative, population level, with frequent repeat measures of HIV status. Historical data provided robust pre-DREAMS measures, and frequent rounds of serological surveys before and after the introduction of DREAMS enabled us to observe trends and identify the timing at which declines began, in relation to DREAMS rollout. DREAMS sought to scale up to district level, and there are very few DREAMS districts with historical HIV surveillance to serve as a baseline for evaluation of DREAMS and track incidence trends in detail. Other evaluations of DREAMS lack biological endpoints or rely on HIV test data from antenatal care clients (thus missing those who are not pregnant or do not seek care) and measure HIV-positive diagnoses at district level as the primary outcome [1]. This can be a reflection of HIV testing reach and positivity rates, rather than a measure of new HIV infections and HIV risk. DREAMS is a very large investment, and direct measurement of HIV incidence at the population level in some settings is warranted to verify if DREAMS achieved its target for HIV incidence reduction. Capitalising on existing research platforms was an efficient way to evaluate this aim.

With large annual survey rounds and high HIV incidence in South Africa, there was sufficient study power to detect a smaller difference than the ambitious DREAMS target of a 40% reduction in HIV incidence, and we found evidence of an average reduction of approximately 25% across the 3 years of DREAMS rollout (2016 to 2018) compared with the years immediately before. (This corresponds to the observed annual reduction (relative to the previous year) of approximately 10% per year.) Although this rate of decline preceded DREAMS introduction, it is encouraging that it continued and did not slow. In Gem, we had sufficient power to detect a minimum change of 40% among 15- to 24-year-old AGYW, in the sensitivity analyses which included residency gaps in the person-years of follow-up and which showed similar results to the original analysis. More person-years of follow-up are needed to provide statistical evidence of a reduction that is less than 40% in this setting, especially the observed 31% reduction in one subgroup: the older cohort of 20- to 24-year-old women.

We proposed a rigorous design in the absence of randomisation. A cluster-randomised trial design was not possible because the priority of the DREAMS Partnership was for rapid rollout of DREAMS investments to geographic areas chosen for their high HIV prevalence, rather than to a randomly selected sample of areas. The absence of a counterfactual (of what would have happened over time in the absence of DREAMS) limits our ability to attribute change to DREAMS or rule out the influence of other interventions and secular trends. It is possible that the sustained decline observed in HIV incidence was due to DREAMS and would not have occurred otherwise. It is arguably more difficult to demonstrate an impact of DREAMS against a background of HIV incidence decline than to do so against a background of stable HIV incidence. There was no evidence that DREAMS reversed the encouraging declines, indicating that it did not disrupt positive trends.

With annual rounds of serosurveillance in South Africa, participation rates in any particular year were relatively low (for example, due to absence from home or study fatigue); however, cumulative rates of participation among eligible individuals increased with time and recruitment opportunities. Overall, about 75% of eligible AGYW contributed person-time during follow-up, which is relatively high for population-based surveillance. In Kenya, the proportion contacted and offered an HIV test was lower, since surveys were done primarily as service provision (home-based testing to link HIV-infected individuals to care), and those with a known HIV positive status did not require a new test. However, the approach to reaching people has remained consistent over time, and the population reached is comparable over time [31]. With less frequent serosurveillance in this setting compared to South Africa, and high mobility

among young adults, there was less opportunity for repeat testing, and, thus, the proportion who contributed person-time to our analyses was lower. While repeated HIV testing via population-based surveillance is considered by many to be the gold standard for measuring HIV incidence trends, such studies are challenged by the systematic exclusion of some subgroups due to noncontact or refusal [32]. The consequence is most likely that absolute levels of HIV incidence in this and other studies may be underestimated, because more mobile groups are disproportionately missed [33]. However, as noted above, in uMkhanyakude, a high proportion of eligible AGYW contributed follow-up time to the analysis, and the proportion contributing to the analysis was fairly consistent over the study period. Also, in previous analyses of participation dynamics in the South Africa surveillance site, the HIV testing rate did not differ substantially by age or sex over time, and the demographic composition (age and sex) of the HIV–negative testers and the HIV cohort remained stable over time [32,34].

Our method of estimating HIV incidence may underestimate HIV incidence rates, if young people are less able to meet the need to have tested twice (for example, if they miss a survey round because they are too young to participate). This is more likely to be the case in Gem than in uMkhanyakude where there were more frequent opportunities to test. While participation rates were lower in uMkhanyakude, previous research from this setting suggests that participation bias can reduce the accuracy with which seroconversions can be dated, undermining validity, but does not introduce a systematic bias. Longer-term follow-up, with the benefit of additional serological data collected after 2019, would increase the accuracy of the "tail-end" of our estimates, since people did not participate in every HIV survey round.

## Conclusions

HIV risk has been persistently high among AGYW in east and southern Africa, and these data offer detailed and encouraging evidence of recent and large declines in 2 settings with an historically high burden of HIV. Frequent rounds of directly observed HIV incidence reveal that the declines predated DREAMS introduction in both settings and thus cannot be attributed to the initial years of DREAMS interventions. They are most likely driven by improvements in HIV services and treatment of HIV–positive individuals, and voluntary male medical circumcision, validating the importance of sustained investment in early diagnosis, treatment, and prevention, for young adult men as well as women. Nevertheless, young women's risk remains high—in absolute levels in South Africa and relative to men of the same age in both settings. HIV incidence reductions are not on track for epidemic control or global targets to end HIV/AIDS as a public health threat by 2030. A complex intervention like DREAMS needs time to embed and strengthen its impact. Time is also needed to measure the true impact of DREAMS as younger adolescent girls age into sexual activity and higher-risk age brackets. Sustained commitment to promoting gender equity through programmes like DREAMS—which offer structural, social, and biomedical support to young women—are needed more than ever as the global COVID-19 pandemic threatens progress in the related goals of HIV prevention and sustainable development.

## Supporting information

**S1 STROBE Checklist. Strengthening the reporting of observational studies in epidemiology (STROBE) checklist.**
(DOCX)

**S1 Table. Mean age of HIV–negative AGYW who are repeat testers (in the HIV incidence cohort) and those who do not have a repeat test, in uMkhanyakude.**
(DOCX)

**S2 Table. Incidence of HIV infection among AGYW in Gem, by age and DREAMS implementation period: sensitivity analysis without residency gaps.**
(DOCX)

**S3 Table. HIV incidence estimates in young men aged 20–29 years by age group and individual year, 2006–2018 in uMkhanyakude.**
(DOCX)

**S4 Table. Incidence of HIV infection among young men in Gem, by age and DREAMS implementation period: sensitivity analysis without residency gaps.**
(DOCX)

## Acknowledgments

We are grateful to the young women and men who volunteered their time for this study. We thank the dedicated data collection teams in Gem and uMkhanyakude, for their tireless efforts to ensure high-quality data. We acknowledge the valuable input of community advisory groups within the study settings and the following individuals, who helped to make this study possible: Gina Dallabetta, Geoff Garnett, and Emilio Emini (BMGF); Janet Saul (CDC); Emily Gutierrez-Zielinski and Mary Mwangi (CDC Kenya); Mary Glenshaw (CDC South Africa); Despoina Xenikaki, Frankie Liew, and Antonio Duran-Aparicio (LSHTM). The team at AHRI would like to thank Dickman Gareta, Tinofa Mutevedzi, Kobus Herbst, Teresa Smit, Nuala McGrath, Frank Tanser, Deenan Pillay, and Till Barnighausen for early support in protocol development. We are grateful to Basia Zaba for facilitating collaborations for this research partnership and offering scientific guidance for the design and analysis.

## Author Contributions

**Conceptualization:** Isolde Birdthistle, Daniel Kwaro, Maryam Shahmanesh, Judith Glynn, Sian Floyd.

**Data curation:** Isolde Birdthistle, Daniel Kwaro, Maryam Shahmanesh, Natsayi Chimbindi, Vivienne Kamire, Jaco Dreyer, Penelope Phillips-Howard, Sian Floyd.

**Formal analysis:** Daniel Kwaro, Kathy Baisley, Sammy Khagayi, Nondumiso Mthiyane, Jaco Dreyer, Sian Floyd.

**Funding acquisition:** Isolde Birdthistle, Daniel Kwaro, Maryam Shahmanesh, Penelope Phillips-Howard, Judith Glynn, Sian Floyd.

**Investigation:** Isolde Birdthistle.

**Methodology:** Isolde Birdthistle, Kathy Baisley, Nondumiso Mthiyane, Jaco Dreyer, Sian Floyd.

**Project administration:** Isolde Birdthistle, Daniel Kwaro, Maryam Shahmanesh, Natsayi Chimbindi, Vivienne Kamire.

**Supervision:** Isolde Birdthistle, Vivienne Kamire, Penelope Phillips-Howard, Judith Glynn, Sian Floyd.

**Writing – original draft:** Isolde Birdthistle, Sian Floyd.

**Writing – review & editing:** Isolde Birdthistle, Maryam Shahmanesh, Kathy Baisley, Natsayi Chimbindi, Annabelle Gourlay, Penelope Phillips-Howard, Judith Glynn, Sian Floyd.

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
