## [Editor Report · Decision Letter 0]

6 Jan 2021

Dear Dr Birdthistle, 

Thank you for submitting your manuscript entitled "Evaluating the impact of DREAMS on HIV incidence among adolescent girls and young women in large population-based cohorts in Kenya and South Africa" for consideration by PLOS Medicine.

Your manuscript has now been evaluated by the PLOS Medicine editorial staff and I am writing to let you know that we would like to send your submission out for external assessment.

Once your full submission is complete, your paper will undergo a series of checks in preparation for full assessment. Once your manuscript has passed all checks it will be sent for external assessment. 

Kind regards,

Richard Turner, PhD

rturner@plos.org

---

## [Decision Letter · Decision Letter 1]

5 Mar 2021

Dear Dr. Birdthistle,

Thank you very much for submitting your manuscript "Evaluating the impact of DREAMS on HIV incidence among adolescent girls and young women in large population-based cohorts in Kenya and South Africa" (PMEDICINE-D-20-06166R1) for consideration at PLOS Medicine. 

Your paper was discussed among the editors and sent to independent reviewers, including a statistical reviewer. The reviews are appended at the bottom of this email and any accompanying reviewer attachments can be seen via the link below:

[LINK]

In light of these reviews, we will not be able to accept the manuscript for publication in the journal in its current form, but we would like to invite you to submit a revised version that addresses the reviewers' and editors' comments fully. You will appreciate that we cannot make a decision about publication until we have seen the revised manuscript and your response, and we expect to seek re-review by one or more of the reviewers. 

We hope to receive your revised manuscript by Mar 26 2021 11:59PM. Please email us (plosmedicine@plos.org) if you have any questions or concerns.

Please let me know if you have any questions, and we look forward to receiving your revised manuscript. 

Sincerely,

Richard Turner, PhD

rturner@plos.org

Please add a colon followed by a study descriptor (e.g., "...: A before-and-after study") to the title. Please remove the word "large". 

As you study the situation in young men in addition to other groups, we suggest amending the title as appropriate. 

Please add a few words to your abstract to describe what DREAMS consists of - we suggest early in the "Methods and findings" subsection, along with the program dates. 

Please avoid "38% lower" and the like where an aRR is quoted. 

Please add a new final sentence to the "Methods and findings" subsection of your abstract, beginning "Study limitations include ..." or similar and quoting 2-3 of the study's main limitations. 

You mention an a priori analysis plan in the Methods section. Please attach the plan as a supplementary file if available, referred to in the main text. Please highlight analyses that were not prespecified. 

Throughout the text, please format reference call-outs as follows: "... HIV risk [12,13]." (noting the absence of spaces within the square brackets). 

In the reference list, please convert all italics into plain text. Where appropriate, please list 6 author names followed by "et al.". 

Please adapt the use of punctuation in lists of author names to journal style (i.e., "Surname Initial(s), Surname Initials, ..., et al."). 

Please remove information on funding from the end of the text. In the event of publication, this information will appear in the article metadata via entries in the submission form. 

Please add a completed checklist for the most appropriate reporting guideline, e.g., STROBE, as a supplementary file, labelled "S1_STROBE_Checklist" or similar and referred to as such in the Methods section. In the checklist, please refer to individual items by section (e.g., "Methods") and paragraph number rather than by line or page numbers, as the latter generally change in the event of publication. 

Comments from the reviewers:

*** Reviewer #1: 

This is a useful study on the impact of DREAMS on HIV incidence among adolescent girls and young women in two communities in Kenya and South Africa. The statistical methods and analyses are relatively straightforward and mostly adequate. However, there are some major issues needing attention.

1) For the sampling of the cohort in uMkhanyakude, South Africa, the consent rates are generally low (40%-60%); for the Gem data in Kenya, the contact rate and also the HIV results rate are both very low (<50%). For such high missing rates in a population study, what's the likely impact on the true estimate of the trends in reduction of HIV rates? Underestimate or overestimate? This needs to be carefully discussed in the limations. For such a high missing rate, the reliability and believeability of results are subject to scrutiny.

2) Study design. This is a before and after cohort design to exam the impact after introduction of the complex intervention of DREAMS on reducing HIV rates over time. However, how can we be certain that the observed reduction in HIV rates is due to DREAMS only? In other words, we need controls/bench marks to show that the reduction in HIV rates is truely because of DREAMS. Do authors have the national HIV rates for South Afric and Kenya over the same time periods? If they are also in decline (e.g. due to national HIV prevention campaign), then there could be contamination effect that needs to be carefully addressed. Or, the HIV rates over time in the neighbouring areas of these two communities? The observed big reduction in HIV rates in young men in the same period doesn't help the arguement. It means the rates declined without DREAMS for young men. Intuitively one would think the rates in young women might decline anyway without DREAMS in the same period. Then, the quesiton is what is the true impact of DREAMS if any. Without addressing all these questions, one can not confirm the impact of DREAMS with confidence.

*** Reviewer #2: 

Thank you for the opportunity to review this manuscript. This is an important and well written manuscript assessing the impact of the DREAMS programme in two sub-Saharan African settings (one in South Africa and one in Kenya) in terms of reducing HIV incidence among young women and men. The piece is strong and utilizes data where serosurveys were available for periods before and after DREAMS implementation. There were many sensitivity analyses/different assessments of time, which could cause concern around multiple hypothesis testing, yet overall the findings were robust: HIV incidence declined over time, but the trend started and did not accelerate during the DREAMS implementation period. There were a few areas that the authors could have provided greater justification for in terms of differences in analytic approach across settings, but overall my largest concern/thought was why didn't the authors use an interrupted time series analysis. It seems like much of what they were trying to do would have been better served by this method? Also, there was a non-insignificant (though understandable) loss to follow-up over time and how those with follow-up incidence data compared to those without within and across time periods was not explored and potentially could be handled to ensure the robustness of the results (for example using inverse probability weighting). Overall, this is important but considerations of these analytic approaches would strengthen the robustness of the findings.

Abstract: ---

Introduction:

The introduction was well written - no comments

Methods:

For the power calculation, it is disorienting (and feels almost deceptive) to give such different power calculations across the two settings. Unless the DREAMS targets were different in each setting (if so this should be clarified), having a 30% reduction and 90% power in one place and 45% reduction and 80% power (or in parentheses 40% reduction & 70% power) is confusing. It seems best to determine the threshold that you think critical- 40% was the DREAMS target so might be logical and be consistent with the difference aimed to detect across settings. 

Similarly, why were the number of imputations different (n=100 vs. n=250) across sites? If there is a reason please provide, otherwise it would seem that these should be congruous.

How was LTFU handled? The authors note that individuals were included if they had 2+ visits, but for those that were not out-migrations (and censored temporarily or permanently), how was follow-up handled. Inverse probability weighting to account for whom follow-up data was and was not available may be used here and could be important depending on the extent to which follow-up occurred.

One of the assumptions of Poisson regression is a steady incidence rate within time periods. Was this assumption assessed? Did the authors consider an interrupted-time series as a potential approach to assess changes in trends over time as related to the DREAMS intervention implementation? These types of quasi-experimental designs are increasingly used.

Please provide a justification within the methods for why different age bands were considered in relation to men across the two settings.

RESULTS

Please describe how those with and without follow-up in both sites were different/similar. 

Also, why were the participants contacted so much lower in Gem (34%) vs. uMkhanyakude (85%)?

There is no accounting for these issues of contact and follow-up and at least for follow-up methods to account for this (at least in part) could be applied.

Per Table 1b, what happened in 2019 that made only 0.7% of testing results available for those eligible for the HIV incidence cohort?

Table 4 - unclear why comparison groups (categories for which incidence is reported) across years differ by age band. 

Discussion

The authors' point in the discussion that interventions may not meet the needs of those at highest risk (such as women engaged in sex work or transactional sex) is important and perhaps could be expounded upon.

The authors should mention as a limitation (and ideally attempt to address) the potential impact of LTFU over intervals on their results.

*** Reviewer #3: 

Thank you for the opportunity to review this article - it is such important research considering the need to prioritize funding for programming in the post-COVID-19 era.

My feedback is two-fold:

big picture:

* the authors do not touch on this but it is a critical question to evaluating impact/ effectiveness: quality of DREAMS implementation. While there is an acknowledgement of the timing of introducing DREAMS (start and end) and how set up may take time, a point in the discussion on how quality and reach of DREAMS may have shaped the results documented.

* where to next? It would be helpful for authors to reflect on the implications of their really impressive research in two key ways: (1) integrating and enhancing data collection in implementation of large-scale programmes - costs, skills, timelines, (2) methodologies of measuring and evaluating impact in scaled up programming. i.e. should there always be a lagged approach to expecting, documenting and analysing for impact?

detailed questions/ items to clarify:

* power calculations - while it makes sense that they vary by site since data collection designs were different, it is unclear why the SA goal was 90% power to detect 30% while Kenya 80% to detect 45%? This is also a bit confusing because in SA the PY included were then much larger than the power calculation.

* air-time - please give amount and who/ where it was sent to (i.e. young adolescent or their caregiver's phone?)

* p-values, please add these throughout to make it easy to follow your interpretation - some of the narrative around the reported values sometimes speaks about impact and at times does seem close to a non-significant. Could the authors please add all p-values and be clear about significance vs interpretation of trends, even when not significant throughout the manuscript. This is quite important because in page 10 the authors report: "but there was a decline among the 20-24 year olds (aRR=0.69, 95%CI 0.53-2.18), albeit with considerable uncertainty around these estimates." but the discussion then speaks to "There is one exception to this finding: among young women aged 20-24 years in Gem, HIV incidence remained stable prior to DREAMS introduction (at circa 1% between 2010-2015), and declined by 31% during the DREAMS implementation period 2016-2019 but with considerable

uncertainty around this estimate." but as I understand the data, this reduction is non significant and with a wide confidence interval. To help all of us reading the manuscript, it would be helpful to set the parameters for defining reduction and the types of reduction (significant/ certain; uncertain) in the methods.

* in the discussion the authors speak to positive causal impact on related outcomes - but the two references are quite broad and dont clarify methods, etc. Could they include some more detail in the discussion until publications are available with the data?

It is really great to see these findings - all of us in the HIV prevention community have been looking forward to this data and learning from DREAMS.

***

[LINK]

---

## [Decision Letter · Decision Letter 2]

17 Sep 2021

Dear Dr. Birdthistle,

Thank you very much for re-submitting your manuscript "Evaluating the impact of DREAMS on HIV incidence among adolescent girls and young women: a population-based cohort study in Kenya and South Africa" (PMEDICINE-D-20-06166R2) for consideration at PLOS Medicine. We apologize for the delay in sending you a response. 

I have discussed the paper with editorial colleagues and it was also seen again by two reviewers. I am pleased to tell you that, provided the remaining editorial and production issues are fully dealt with, we expect to be able to accept the paper for publication in the journal.

[LINK]

Please let me know if you have any questions, and we look forward to receiving the revised manuscript.

Sincerely,

Richard Turner, PhD

rturner@plos.org

Requests from Editors:

Are you able to add additional information to the data statement to indicate the folder or files in the two repositories that contain the relevant study data?

Early in the "Methods and findings" subsection of your abstract, please add a sentence, say, to describe the nature and dates of the surveillance surveys.

Again in the abstract, prior to quoting the findings on HIV incidence, please quote some relevant numbers from table 1, which might include the range of survey cohort sizes, the range of proportions contacted, and the range of proportions consented. 

In the abstract, when comparing numbers for HIV incidence, please convert instances of "from X new infections ... to Y" to "X as compared with ... Y" or similar.

Please make similar changes in the Results section (main text). 

Please restructure the "Author summary", as the first two points of the "What do these findings mean?" subsection belong in the previous subsection ("What did the researchers do and find?"). Please aim for 2-4 points in each subsection, overall. 

In the Methods section (main text), you mention both an a priori analysis plan and study protocol. Are you able to provide one or both documents as attachments? If so, please refer to these in the text. 

We suggest avoiding the apostrophe in "DREAMS' introduction" and similar phrases, which will be difficult to make consistent, instead writing "introduction of DREAMS" and the like. 

Throughout the text, please style reference call-outs as follows: "... early 2016 [5,6]." (noting the absence of spaces within the square brackets). 

In the reference list, please use the journal name abbreviation "PLoS ONE" consistently.

Noting reference 2 and others, please add abstract numbers to citations of conference presentations, where available.

Please update reference 18: it may be necessary to remove this reference if not available as a preprint or "in press". 

Comments from Reviewers:

*** Reviewer #1: 

Many thanks authors for their great effort to improve the manuscript. The authors have addressed my comments professionally. I am satisfied with the response and revision. No further issues needing attention.

*** Reviewer #3: 

Thank you for this revised manuscript. Overall, this version addresses many of the comments and feedback. However, the reporting of HR throughout remains inconsistent with uncertain language around non-significant results. Could the authors be very clear in the methods (analyses) how they are interpreting results and 95% CI, then use that throughout. Include all p-values (not only some) and then be clear about what is interpreting the "direction" or "magnitude" of a significant or non-significant result. Thank you.

***

[LINK]

---

## [Editor Report · Decision Letter 3]

5 Oct 2021

Dear Dr Birdthistle, 

On behalf of my colleagues and the Academic Editor, Dr Newell, I am pleased to inform you that we have agreed to publish your manuscript "Evaluating the impact of DREAMS on HIV incidence among adolescent girls and young women: a population-based cohort study in Kenya and South Africa" (PMEDICINE-D-20-06166R3) in PLOS Medicine.

Prior to final acceptance, where available please add further details to the abstract citations that are included, e.g., references 24 & 25.

PRESS

Sincerely, 

Richard Turner, PhD 

rturner@plos.org